# Central Modulation of Selective Sphingosine-1-Phosphate Receptor 1 Ameliorates Experimental Multiple Sclerosis

**DOI:** 10.3390/cells9051290

**Published:** 2020-05-22

**Authors:** Alessandra Musella, Antonietta Gentile, Livia Guadalupi, Francesca Romana Rizzo, Francesca De Vito, Diego Fresegna, Antonio Bruno, Ettore Dolcetti, Valentina Vanni, Laura Vitiello, Silvia Bullitta, Krizia Sanna, Silvia Caioli, Sara Balletta, Monica Nencini, Fabio Buttari, Mario Stampanoni Bassi, Diego Centonze, Georgia Mandolesi

**Affiliations:** 1Department of Human Sciences and Quality of Life Promotion, University of Rome San Raffaele, 00166 Rome, Italy; alessandra.musella@sanraffaele.it (A.M.); georgia.mandolesi@sanraffaele.it (G.M.); 2IRCCS San Raffaele Pisana, 00166 Rome, Italy; gntnnt01@uniroma2.it (A.G.); diego.fresegna@gmail.com (D.F.); valentina_vanni@hotmail.it (V.V.); laura.vitiello@sanraffaele.it (L.V.); Silvia.Bullitta@uniroma2.it (S.B.); monicanencini@hotmail.com (M.N.); 3Department of Systems Medicine, University of Rome Tor Vergata, 00133 Rome, Italy; livia.guadalupi@gmail.com (L.G.); f.rizzo@med.uniroma2.it (F.R.R.); antonio.bruno91@yahoo.it (A.B.); ettoredolcetti@hotmail.it (E.D.); krizia.sanna@live.it (K.S.); silviacaioli@yahoo.it (S.C.); balletta.sara@gmail.com (S.B.); 4Unit of Neurology, IRCCS Neuromed, 86077 Pozzilli (IS), Italy; f.devito.molbio@gmail.com (F.D.V.); fabio.buttari@gmail.com (F.B.); mario_sb@hotmail.it (M.S.B.)

**Keywords:** sphingosine-1-phosphate receptors, glutamate synaptic dysfunction, microglia, T lymphocytes, experimental autoimmune encephalomyelitis (EAE), pro-inflammatory cytokines, neuroinflammation, ozanimod, AUY954, A971432, S1P_1_, S1P_5_

## Abstract

Future treatments of multiple sclerosis (MS), a chronic autoimmune neurodegenerative disease of the central nervous system (CNS), aim for simultaneous early targeting of peripheral immune function and neuroinflammation. Sphingosine-1-phosphate (S1P) receptor modulators are among the most promising drugs with both “immunological” and “non-immunological” actions. Selective S1P receptor modulators have been recently approved for MS and shown clinical efficacy in its mouse model, the experimental autoimmune encephalomyelitis (EAE). Here, we investigated the anti-inflammatory/neuroprotective effects of ozanimod (RPC1063), a S1P_1/5_ modulator recently approved in the United States for the treatment of MS, by performing ex vivo studies in EAE brain. Electrophysiological experiments, supported by molecular and immunofluorescence analysis, revealed that ozanimod was able to dampen the EAE glutamatergic synaptic alterations, through attenuation of local inflammatory response driven by activated microglia and infiltrating T cells, the main CNS-cellular players of EAE synaptopathy. Electrophysiological studies with selective S1P_1_ (AUY954) and S1P_5_ (A971432) agonists suggested that S1P_1_ modulation is the main driver of the anti-excitotoxic activity mediated by ozanimod. Accordingly, in vivo intra-cerebroventricular treatment of EAE mice with AUY954 ameliorated clinical disability. Altogether these results strengthened the relevance of S1P_1_ agonists as immunomodulatory and neuroprotective drugs for MS therapy.

## 1. Introduction

Multiple sclerosis (MS) is an autoimmune disease extensively reported to be triggered by myelin-targeted T-cell infiltration into the central nervous system (CNS) and by B-cell autoantibodies. The resulting brain inflammation is regarded as the main cause of both demyelination and neurodegeneration [1,2,3,4,5,6]. Neurodegeneration combined with a lack of repair is increasingly recognized to contribute to disease progression and to occur independently of focal white matter damage [5]. Recently, several preclinical and clinical studies have demonstrated that a marked and early imbalance between glutamate excitatory and GABA inhibitory transmission affects the brain of both MS patients and mice with experimental autoimmune encephalomyelitis (EAE), a mouse model of MS [5,7,8,9,10]. Of note, pro-inflammatory cytokines released by activated microglia and lymphocytes in both EAE and MS have been shown to exert direct synaptopathic effects especially on glutamatergic transmission [11].

Together with the loss of synapses, these synaptic alterations lead to a diffuse “synaptopathy” that is driven by inflammation and, in turn, causes excitotoxicity, a well-known mechanism of neurodegeneration. Therapeutic strategies to target synaptopathy are particularly appealing, because —unlike loss of neurons—synaptic dysfunction and loss of synapses are reversible processes. In this respect, several studies in EAE have shown that disease modifying therapies (DMTs) in use for MS, beyond their peripheral immunomodulation, can target the inflammatory synaptopathy, providing neuroprotection [12,13,14,15].

Developing neuroprotective and neurorepair therapeutic strategies is required for the future treatment of MS. A simultaneous and early targeting of peripheral immune cell function and of CNS-intrinsic inflammation, potentially through combinatorial therapies designed to modulate these two immunological arms of the disease, along with the provision of neuroprotective or neuroregenerative drugs, are challenging therapeutic goals. Promising molecules with both immunological and non-immunological actions are represented by modulators of the family of sphingosine-1-phosphate (S1P) receptors [16]. S1P is a powerful bioactive sphingolipid that activates S1P receptors in an autocrine and/or paracrine way [17]. The five receptor subtypes that have been identified, S1P_1_, S1P_2_, S1P_3_, S1P_4_, and S1P_5_, are G-protein-coupled receptors that are almost ubiquitously expressed and play important roles in cell survival, growth, and differentiation in many cell types, including cells of the immune and central nervous system [18]. Modulation of lymphocyte trafficking, mainly mediated by S1P_1_, is the main mechanism responsible for the beneficial effects of S1P drugs, such as fingolimod (FTY720), the first nonselective S1P receptor agonist approved for relapsing remitting (RRMS) patients [19,20] and siponimod (BAF312), a S1P_1_ and S1P_5_-selective fingolimod-congener, one of the few MS drugs approved for secondary progressive MS (SPMS) [21]. However, the relevant observation of a reduced brain volume loss in fingolimod-treated RRMS patients [22], and the successful reduction of neurological damage accumulation in siponimod-treated SPMS patients, highlighted a “non-immunological” effect due to modulation of S1P receptors in CNS resident cells, including neurons, astrocytes, oligodendrocytes, and microglia. Studies in experimental MS, especially in the EAE model, have attempted to define the molecular mechanisms underlying these effects, highlighting the involvement of S1P_1_ or S1P_5_ specifically expressed on astroglia and oligodendrocytes, respectively [23,24,25,26,27]. In our previous paper [13], we performed a long-lasting, preventive, and central delivery of siponimod in EAE mice and we observed a late amelioration of clinical disability. Such a beneficial effect was accompanied by a reduced astrogliosis and microgliosis in the EAE striatum, a brain region particularly affected in EAE, and by neuroprotective action on GABA-ergic neurons, likely because of a S1P_1_-mediated reduction of the central inflammation [13]. Despite a reduced microgliosis, we could not detect any amelioration of EAE glutamate-mediated excitotoxicity [5,28], suggesting a specificity of siponimod action in this experimental condition. These results also raised the possibility that different S1P modulators could elicit specific anti-synaptotoxic effects.

In the present study, by means of ex vivo and in vitro experiments we investigated the anti-excitotoxic activity of ozanimod (RPC1063), an MS oral S1P_1_/S1P_5_ modulator [29], on glutamatergic transmission alterations in EAE striatum. Given the safety and efficacy profile, ozanimod has recently been approved in the United States for clinically isolated syndrome, RRMS, and active SPMS. Ozanimod also does not require first dose heart rate monitoring as with fingolimod, nor genetic CYP2C9 screening as with siponimod [30]. Furthermore, by using ex vivo and/or central in vivo approaches, we addressed the potential neuroprotective effects of selective S1P_1_ and S1P_5_ agonists on EAE mice, in order to discriminate the different involvement of these S1P receptor subtypes in ameliorating synaptic alterations and clinical disability.

## 2. Methods

### 2.1. Drug Formulation

Based on EC_50_ values [29,31,32] and to achieve selectivity for S1P receptors, for ex vivo experiments we selected the following concentration in nM: ozanimod (1000), AUY954 (300); A971432 (200). Drugs were dissolved in DMSO (0.1–0.25% final concentration) for in vitro experimentation. For in vivo experiments two AUY954 dosages were tested: 2.7 and 0.55 μg/day; the drug was dissolved in a solution containing 10% Solutol/Kolliphor HS15 (BASF Pharma Solutions), final pH range between 6 and 7.

### 2.2. Mice

Animals employed in this study were 6–8-week-old C57BL/6 female mice, obtained from Charles-River (Milan, Italy). Mice were housed under constant conditions in an animal facility with a regular 12 h light/dark cycle. Food and water were supplied *ad libitum*. All the efforts were made to minimize the number of animals used and their suffering. In particular, when animals experienced hindlimb weakness, moistened food and water were made easily accessible to the animals on the cage floor. Mice with hindlimb paresis received glucose solution by subcutaneous injection or food by gavage during the entire procedure. In the rare presence of a tetraparalyzed animal, mice were sacrificed. Minipump-implanted mice were housed in individual cages endowed with special bedding (TEK-Fresch, Envigo, Casatenovo (LC), Italy) in order to avoid skin infections around the surgical scar. Animal experiments were performed according to the Internal Institutional Review Committee, the European Directive 2010/63/EU and the European Recommendations 526/2007, and the Italian D.Lgs 26/2014.

### 2.3. EAE Model

Chronic-progressive EAE was induced as previously described [8,33]. Six-eight weeks old C57BL/6 female mice were active immunized with an emulsion of mouse myelin oligodendrocyte glycoprotein peptide 35–55 (MOG35–55, 85% purity; Espikem, Prato, Italy) in Complete Freund’s Adjuvant (CFA; Difco, Los Altos, CA, USA), followed by intravenous administration of pertussis toxin (500 ng; Merck, Milan, Italy) on the day of immunization and two days post immunization (dpi). Animals were scored daily for clinical symptoms of EAE according to the following scale: 0 no clinical signs; 1 flaccid tail; 2 hindlimb weakness; 3 hindlimb paresis; 4 tetraparalysis; and 5 death due to EAE; intermediate clinical signs were scored by adding 0.5.

### 2.4. Minipump and Surgery

One week before immunization, mice were implanted with subcutaneous osmotic minipumps allowing continuous intracerebroventricular (icv) infusion of either vehicle or AUY954 for 4 weeks as described in [8,34].

### 2.5. Ex Vivo Experiments

EAE mice (21–25 dpi) were killed by cervical dislocation and corticostriatal slices (200 μm) were prepared from fresh tissue blocks of the brain with the use of a vibratome (Leica VT1200 S) [28].

For molecular biology experiments, the slices were incubated for 2 h with ozanimod (1000 nM) or vehicle (DMSO 0.1% final concentration) in a chamber containing oxygenated artificial cerebrospinal fluid (ACSF). After incubation, the cortex was removed and the striatum was stored at −80 °C until use.

For the ex vivo electrophysiological experiments, fresh EAE striatal slices were incubated with AUY954 (300 nM), A971432 (200 nM), or ozanimod (1000 nM) for 2 h. A one-hour incubation of T lymphocytes isolated from EAE mice and treated with ozanimod (overnight) was performed before electrophysiological recordings when specified.

### 2.6. In Vitro Experiments

The BV2 immortalized murine microglial cell line was cultured as previously described [13]. BV2 cells were maintained in a humidified incubator with 5% CO_2_ and cultured in DMEM supplemented with 5% FBS, 100 U/mL penicillin, and 100 mg/mL streptomycin. BV2 cells were pre-treated for 1 h with vehicle (DMSO) or ozanimod (1000 nM) and then in vitro activated for 2 h with a Mix of Th1-specific proinflammatory cytokines: 100 U/mL IL-1β (Euroclone, Milan, Italy), 200 U/mL tumor necrosis factor (TNF, Miltenyi Biotec, Bologna, Italy), and 500 U/mL interferon γ (IFNγ, Becton Dickinson, Milan, Italy) [28]

### 2.7. Electrophysiology

Mice were killed by cervical dislocation, and corticostriatal coronal slices (200 μm) were prepared from fresh tissue blocks of the brain with the use of a vibratome. A single slice was then transferred to a recording chamber and submerged in a continuously flowing ACSF (34 °C, 2–3 mL/min) gassed with 95% O_2_–5% CO_2_. The composition of the control ACSF was (in mM): 126 NaCl, 2.5 KCl, 1.2 MgCl_2_, 1.2 NaH_2_PO_4_, 2.4 CaCl_2_, 11 glucose, 25 NaHCO_3_. To study spontaneous glutamate-mediated excitatory postsynaptic currents (sEPSCs), the recording pipettes were filled with internal solution of the following composition (mM): K^+^-gluconate (125), NaCl (10), CaCl_2_ (1.0), MgCl_2_ (2.0), 1,2-bis (2-aminophenoxy) ethane-*N*,*N*,*N*,*N*-tetra acetic acid (BAPTA; 0.5), HEPES (19), GTP; (0.3), Mg-ATP; (1.0), adjusted to pH 7.3 with KOH. Bicuculline (10 µM) was added to the external solution to block GABA_A_-mediated transmission. The detection threshold of sEPSCs was set at twice the baseline noise. Offline analysis was performed on spontaneous synaptic events recorded during fixed time epochs (1–2 min, three to five samplings), sampled every 5 or 10 min. Only cells that exhibited stable frequencies in control (less than 20% changes during the control samplings) were used for analysis. For kinetic analysis, events with peak amplitude between 10 and 50 pA were grouped, aligned by half-rise time, normalized by peak amplitude, and averaged to obtain rise times and decay times.

### 2.8. Real Time PCR (qPCR)

Total RNA was extracted from treated BV2 cells and EAE striatal slices according to the standard miRNeasy Micro kit protocol (Qiagen). The RNA quantity and purity were analyzed with the Nanodrop 1000 spectrophotometer (Thermo Scientific). The quality of RNA was assessed by visual inspection of the agarose gel electrophoresis images. Next, 700–900 ng of total RNA was reverse-transcribed using high-capacity cDNA reverse transcription kit (Applied Biosystem) according to the manufacturer’s instructions and 3–24 ng of cDNA were amplified in triplicate using the Applied Biosystem 7900HT Fast Real Time PCR system. mRNA relative quantification was performed using the comparative cycle threshold (2^−ΔΔCt^) method. β-actin was used as endogenous control. For the mRNA quantification of cytokines, Tgfb1, RANTES, and iNOS and FIZZ1, SensiMix II Probe Hi-Rox Kit (Bioline; Meridian Life Science) and the following TaqMan gene expression assays were used: IL-1b ID: Mm00434228_m1; Tnf ID: Mm00443258_m1; IL6 ID: Mm00446190_m1; Il10 ID: Mm01288386_m1; Tgfb1 ID: Mm01178820_m1; Ccl5 (coding for RANTES) ID:Mm01302427_m1; Nos2 (coding for iNOS) ID: Mm00440502_m1; Retnla (coding for FIZZ1) ID: Mm00445109_m1; Actb ID: Mm00607939_s1. SensiMix SYBR Hi-Rox Kit (Bioline) was utilized for the quantification of mRNA coding for ionized binding protein type-1 (IBA1) by using the following primers: Aif1 mRNA coding for IBA1 (NM_019467): forward GACAGACTGCCAGCCTAAGACAA, reverse CATTCGCTTCAAGGACATAATATCG; β-actin (NM_007393): forward CCTAGCACCATGAAGATCAAGATCA, reverse AAGCCATGCCAATGTTGTCTCT.

### 2.9. Immunohistochemistry and Confocal Microscopy

Striatal slices (200 µm) cut from the brain of 21–25 dpi EAE mice were incubated in oxygenated ACSF in the presence of vehicle (DMSO 0.1% final concentration) or ozanimod (1000 nM) for 2 h. Slices were fixed in 4% PFA and equilibrated with 30% sucrose before cutting 30 μm slices, to perform immunohistochemistry and confocal microscopy experiments [8]. Sections were permeabilized in PBS with Triton-X 0.25% (TPBS). All following incubations were performed in TPBS. Sections were pre-incubated with 10% normal donkey serum solution for 1 h at room temperature and incubated with the appropriate mix of following primary antibodies: rabbit anti-Iba1 (1:750, Wako, Milan, Italy), goat anti-IL-1β (1:300, R&D System, Milan, Italy), mouse anti-TNF (1:1500, Abcam, Milan, Italy) overnight at 4 °C. After three washes, 10 min each, sections were incubated with the secondary antibody Alexa-488 conjugated donkey anti-Rabbit (1:200, Invitrogen, Milan, Italy); Cy3-conjugated donkey anti-mouse or anti-goat (1:200, Jackson, Milan, Italy) for 2 h at RT, rinsed and DAPI counterstained. Sections were mounted with Vecta-shield (Vector Labs, Milan, Italy) on poly-l-lysine-coated slides, air-dried and coverslipped. Images from immunolabeled samples were acquired using a model LSM7 Zeiss confocal laser-scanner microscope with 20× objective (zoom 1.5×). The images had a pixel resolution of 1024 × 1024. The confocal pinhole was kept at 1.0, the gain and the offset were lowered to prevent saturation in the brightest signals, and sequential scanning for each channel was performed. Images were exported in Tiff format and adjusted for brightness and contrast as needed using ImageJ software. Acquisitions were made on three slices cut from three EAE mice and incubated with ozanimod or vehicle.

### 2.10. Murine CD3+ Cell Isolation

Mice were sacrificed at 15–25 dpi through cervical dislocation and the spleens were quickly removed and stored in sterile phosphate-buffered saline (PBS). After mechanical dissociation of the tissue, the cell suspension was passed through a 40-µm cell strainer (BD Biosciences, San Jose, CA, USA) to remove cell debris and centrifuged. The cell suspension obtained was subjected to magnetic cell sorting separation (CD3 microbeads kit; Miltenyi Biotec) to obtain a pure lymphocyte population. T cells were co-incubated overnight with ozanimod (1000 nM). About 5 × 10^3^ pure T cells were incubated with striatal slices for 30–60 min, in a total volume of 1 mL of oxygenated ACSF before the electrophysiological recordings.

### 2.11. T-Cell Absolute Count

T-cell absolute count was performed on blood samples kept from the mandibular vein of the mouse. To evaluate T lymphocyte number, we used an anti-mouse CD3 PE conjugated antibody (BD Biosciences, San Jose, CA, USA, clone 145–2C11). At predetermined optimal concentrations, 100 μL of blood was stained by incubation with the antibody. Fifty microliters of Count Bright Absolute Counting Beads (Molecular Probes, Milan, Italy) were added, and erythrocytes were lysed using ACK solution (Lonza Bio Whittaker, Walkersville, MD, USA), according to standard protocols. Cell suspensions were acquired and analyzed on LSR Fortessa™ X20 SO Cell Analyzer (BD Biosciences). By comparing the ratio of bead events to cell events, absolute numbers of cells in the sample were calculated. In particular, the formula used was: (number of cells counted/number of beads counted) × (lot-specific number of beads/sample volume).

### 2.12. Statistical Analysis

For each type of experiment, at least three mice of each group were used. Data are presented as means ± SEM. The significance level was established at *p* < 0.05. Statistical analysis was performed using paired or unpaired Student’s *t*-test. Multiple comparisons were analyzed by one-way ANOVA, followed by Tukey’s HSD. Differences between groups in clinical score analysis were tested by Mann-Whitney test. Linear regression test was used to calculate correlation between clinical score and T cell counts. Throughout the text experiments “*N*” refers to the number of animals, while “*n*” refers to the number of cells for electrophysiological experiments and to the number of slices or biological samples for molecular biology experiments.

## 3. Results

### 3.1. Ex Vivo Ozanimod Treatment Restores Normal Glutamatergic Transmission in EAE Striatum

We first explored a potential anti-excitotoxic effect of the S1P_1/5_ modulator ozanimod. To this aim, we induced MOG_(35–55)_ EAE in a group of mice and performed ex vivo treatments and glutamatergic transmission recording in corticostriatal slices derived from EAE mice (20–25 dpi) [28]. In details, we performed ex vivo incubation of EAE corticostriatal slices with ozanimod (1000 nM) or vehicle for two hours and performed whole-cell voltage-clamp recordings from medium spiny neurons (MSNs) to evaluate both the duration and the frequency of sEPSCs. These post- and pre-synaptic parameters, respectively, are typically enhanced in EAE condition relative to control healthy mice [28].

Ozanimod reduced the duration (decay time and half width) of glutamate-mediated spontaneous currents compared to EAE-vehicle, reaching values similar to healthy mice (dashed lines) (20–25 dpi, EAE *n* = 8, EAE + ozanimod *n* = 8: EAE vs. EAE + ozanimod decay time *p* < 0.05, half width *p* < 0.01; EAE + ozanimod vs. control *p* > 0.05 for both half width and decay time; EAE vs. control *p* < 0.01 for both half width and decay time) (Figure 1A–C). Moreover, we investigated the effect of ozanimod on other sEPSC parameters, demonstrating that ozanimod treatment also recovered the frequency without affecting the amplitude (Figure 1A,D,E).

These results highlight a direct anti-excitotoxic impact of ozanimod on EAE corticostriatal slices.

### 3.2. Ozanimod Treatment Exerts an Anti-Inflammatory Action on EAE Striatum and on Activated Microglial Cell Line

The observed beneficial effect of ozanimod on glutamatergic dysfunction in EAE slices suggests its potential local immunomodulatory activity. Of note, microglia, which mainly express S1P_1_, are regarded as the main source of inflammatory mediators that contribute to synaptopathy in the EAE/MS brains [8,35]. Thus, by qPCR we analyzed mRNA levels of specific markers of microglia activation and inflammation in corticostriatal slices treated with ozanimod (as described for electrophysiological experiments).

First, we quantified the expression levels of microglia-specific transcripts coding for the binding adaptor molecule 1 (IBA-1) and M1- and M2-like markers, the inducible nitric oxide synthetase (iNOS) and FIZZ1, also known as resistin like alpha (Retnla), respectively. We observed that the M2-like marker FIZZ1 was significantly up-regulated following 2 h of ozanimod incubation (EAE 20–25 dpi, EAE *n* = 4 slices, EAE + ozanimod *n* = 5 slices; EAE vs. EAE + ozanimod *p* < 0.001), while the mRNA levels of IBA-1 and iNOS did not significantly change (EAE vs. EAE + ozanimod *p* > 0.05; Figure 2A).

Next, we investigated the expression levels of several pro-(TNF, IL-1β, and IL-6) and anti-(TGF1β, IL-10) inflammatory cytokines and the chemokine RANTES as a measure of the immunomodulatory activity of ozanimod. As shown in Figure 2B, treatment with ozanimod (1000 nM) downregulated TNF and IL-1β mRNAs (EAE 20–25 dpi, EAE *n* = 11 slices, EAE + ozanimod *n* = 12 slices; EAE vs. EAE + ozanimod TNF: *p* < 0.001; IL-1β: *p* < 0.05).

To support qPCR data, we performed immunofluorescence experiments on EAE striatal slices incubated with ozanimod and vehicle, focusing on TNF and IL-1β, which have been clearly shown to alter the glutamatergic transmission during EAE [8,28]. By means of confocal imaging we confirmed the remarkable effect of ozanimod on IL-1β, while the effect on TNF protein was less evident (Figure 3A,B). Indeed, we observed that ozanimod strongly attenuated IL-1β staining (Figure 3A, red) in lesioned area of EAE striatal slices, characterized by an intense labeling of the microglia/macrophage activation marker Iba1 (Figure 3A, green; counterstaining with dapi-cyan). In contrast, TNF immunolabeling was slightly reduced in ozanimod treated slices (Figure 3B, red).

Finally, to further support a direct neuromodulatory effect of ozanimod on microglia cells, BV2 cells were in vitro treated with ozanimod and activated with a mix of Th1 cytokines known to mimic EAE condition [28]. After treatment, activated BV2 cells and the relative controls were processed for qPCR analysis to quantify mRNAs of IL-6, IL-1β, TNF, RANTES/CCL5, and IL-10. As shown in Figure 4, ozanimod treatment induced a downregulation of IL-6, RANTES/CCL5, and TNF mRNAs (n = 3 biological samples slices per condition, *p* < 0.05), with no effects on IL-1β and the anti-inflammatory cytokine IL-10 mRNAs in comparison to Th1 mix control condition (n = 3 biological samples slices for each treatment, *p* > 0.05; data not shown).

### 3.3. Ozanimod Pre-Treatment of EAE T Lymphocytes Rescues Striatal Glutamatergic Alterations In Ex Vivo EAE Model

Infiltrating T cells are another important source of cytokines in the inflamed EAE and MS brains [6,36]. Noteworthy, we previously reported that T lymphocytes isolated from the spleen of EAE mice induced a TNF-dependent enhancement of the kinetic properties of the sEPSCs when incubated for 1–2 h with healthy corticostriatal slices, mimicking EAE condition [28]. Therefore, we addressed the hypothesis that the neuroprotective effect of ozanimod could be mediated by its immunomodulatory activity on T lymphocytes, beside its well-known peripheral effect on CNS-accessing T cells. To this aim, we pre-incubated EAE lymphocytes with ozanimod overnight (1000 nM) and then we assessed their synaptotoxic effect on striatal slices of healthy mice by recording striatal glutamatergic currents. Decay time and half width parameters of sEPSCs were significantly reduced following incubation with ozanimod-treated EAE lymphocytes compared to untreated cells (ozanimod-treated EAE T cells *n* = 16, EAE T cells *n* = 5; unpaired *t*-test, decay time: ozanimod-treated EAE T cells vs. EAE T cells *p* < 0.01; half width ozanimod-treated EAE T cells vs. EAE T cells *p* < 0.01; Figure 5).

This experimental strategy that allows the investigation of a direct synaptotoxic role for infiltrating lymphocytes, together with the evidence that T cells predominantly express S1P_1_ reinforce the central neuroprotective action of ozanimod mediated by S1P_1_ modulation.

### 3.4. Selective Agonists of Central S1P_1_ and S1P_5_ Differently Modulate EAE Striatal Glutamatergic Alterations

The above data point to a major role of S1P_1_ in mediating the anti-synaptotoxic effects of ozanimod in EAE mice. In order to assess whether ozanimod agonism on S1P_1_ could account for such beneficial action, we tested the effects of AUY954 and A971432, S1P_1_, and S1P_5_ agonist respectively, on EAE corticostriatal synaptic transmission. To this aim, we incubated EAE striatal slices for 2 h ex vivo with AUY954 (300 nM), A971432 (200 nM), or vehicle and performed whole-cell voltage-clamp recordings from MSNs to evaluate both pre- and post- synaptic parameters of sEPSCs. The statistical analysis of sEPSC kinetics, specifically decay time and half width, showed that both AUY954 and A971432 reduced, at least in part, the duration of the sEPSCs that is significantly increased in EAE striatum (Figure 6). Specifically, AUY954 was the most effective drug modulating both half width and decay time, reaching values similar to those of the control condition (EAE *n* = 15, EAE + AUY *n* = 7, control mice *n* = 9; decay time: EAE vs. control *p* < 0.05; EAE vs. EAE + AUY *p* < 0.01; half width: EAE vs. control *p* < 0.001; EAE vs. EAE + AUY *p* < 0.01; Figure 6). Treatment with A971432, instead, had a minor effect, as showed by the incomplete recovery of the decay time (EAE + A971432 *n* = 12; EAE vs. EAE + A971432: decay time *p*> 0.05; half width *p*< 0.05; Figure 6).

The frequency and the amplitude of sEPSCs were unchanged under both treatments with respect to EAE-vehicle condition (EAE *n* = 15, EAE + AUY *n* = 7, EAE + A971432 *n* = 12; control mice *n* = 9; *p*> 0.05 vs. EAE mice, data not shown).

These results show that a local and brief S1P_1_ modulation ameliorates EAE striatal synaptic dysfunction more efficiently than S1P_5_ agonist treatment.

### 3.5. In Vivo Treatment with S1P_1_ Selective Agonist Ameliorates EAE Disease

The next experiments aimed at investigating a direct neuroprotective effect of S1P_1_ modulator in vivo in the EAE model. We preventively treated EAE mice with two different doses of AUY954 (2.7 µg/day and 0.55 µg/day) and vehicle, by means of a continuous icv infusion for four weeks. The higher dose was selected based on the maximum concentration available for this experimental condition. Since the effect on clinical score between high and low dose of AUY954 was similar (data not shown), the data were gathered together. We observed that daily central treatments with the drug ameliorated the clinical score of the disease (EAE-AUY954 icv *N* = 15, EAE-vhl *N* = 6; Mann-Whitney test *p* < 0.05; Figure 7A).

To address the possibility that the amelioration of the clinical score was mediated by a peripheral effect of the drug on lymphocytes, we counted the total number of CD3+ cells in peripheral blood samples of EAE mice that received the two different dosages of AUY954 or vehicle. As expected, EAE induced a significant increase in T lymphocytes, compared to healthy mice (*N* = 4, controls *N* = 3; *p* < 0.05; Figure 7B). Interestingly, irrespective of the dose administered, we did not observe any significant drop in T lymphocyte counts in blood samples of AUY954-treated mice compared to EAE-vehicle mice (*N* = 6, dosage 0.55 µg/day; *N* = 5, 2.7 µg/day; EAE-high and EAE-low *p* < 0.05 compared to control; Figure 7B). Of note, AUY954-treated EAE mice with zero score showed similar CD3+ counts in comparison to EAE-vhl sick mice (mean score 2.5). In accordance, there was no correlation between clinical score and T cell counts in AUY954-treated mice (r^2^ = 0.06; Figure 7C), further supporting the idea that the beneficial effect of AUY954 treatment on clinical disability was due to a central effect of the drug.

## 4. Discussion

In the present study, we investigated the modulatory effects of ozanimod on inflammatory glutamate-mediated excitotoxicity, a pathological feature of EAE and MS brains and we evaluated the different involvement of sphingosine receptor subtypes (S1P_1_ and S1P_5_) in the EAE model. Two main results emerged: first, the S1P_1_/S1P_5_ modulator ozanimod has central neuroprotective effects likely mediated by an action on microglia cells and infiltrating lymphocytes, resulting in a reduced release of proinflammatory cytokines, the main players of inflammatory synaptopathy. Second, the central delivery of a selective S1P_1_ modulator showed neuroprotective effects, in terms of both EAE clinical score and inflammatory synaptopathy, suggesting a primary involvement of this receptor subtype in ozanimod-induced neuroprotection also in MS.

Glutamate-mediated excitotoxicity is increasingly regarded as a relevant pathogenic mechanism of neurodegeneration in MS [37,38]. Importantly, long-lasting potentiation of the glutamatergic transmission, meaning increased glutamate release from presynaptic terminals and prolonged post-synaptic action of the neurotransmitter, contributes to dendritic spine degeneration and neuronal loss in EAE [2]. In the last years, the recognition of excitotoxicity as a pathogenic mechanism in MS has brought to light a novel therapeutic target, in which the S1P receptor family may play a role. In this respect, fingolimod, the first S1P modulator developed for MS treatment, has been shown to exert its therapeutic actions via a multimodal mechanism, which includes not only the expected T cell retention in lymph nodes [39], but also an anti-inflammatory and anti-excitotoxic action in the CNS. In particular, in EAE mice, oral fingolimod was able to restore presynaptic and postsynaptic alterations of glutamatergic transmission and to promote the recovery of dendritic spines [12], probably owing to its ability to suppress T cell infiltration into the brain and to dampen microglia activation and astrogliosis [23,24]. More interestingly, fingolimod was found to reduce glutamate-mediated intracortical excitability measured by paired-pulse TMS in patients with RRMS [40]. Noteworthy, the anti-excitotoxic activity of fingolimod has been proven in models of “pure” excitotoxicity. Fingolimod was shown to protect against excitotoxic insult in cortical neuronal cultures and organotypic slices [25,26,27], as well as in the in vivo models of excitotoxicity induced by kainic acid (KA) or glutamate [26,27].

Fingolimod is a nonselective S1P agonist, showing affinity for S1P_1,3–5_, which does not allow to discriminate which receptor subtype is involved in such synaptic activity. In the search for S1P receptor modulators with direct CNS activity, drug design studies have focused on S1P_1_ and S1P_5_ agonists, like siponimod that has shown remarkable neuroprotective effects in human and experimental MS [13,41]. In particular, icv delivery of siponimod in EAE mice improved clinical disability during the late phase of the disease and ameliorated the GABAergic defects of EAE striatum but not the glutamatergic counterpart [13]. Of note, an enhanced survival of GABA-ergic neurons as well as a reduced neuroinflammation accounted for such synaptic effect. The neuroprotective effect of siponimod might involve the pro-survival signaling mediated by brain-derived neurotrophic factor (BDNF), as observed with fingolimod [42]. Recently, activation of AKT and ERK kinases as well as an increase of BDNF levels have been associated with a neuroprotective effect mediated by selective S1P_5_ agonists in other neurological disease [42,43,44].

To better clarify the putative mechanisms underlying the anti-synaptopathic activity of S1P_1/5_ agonism at glutamatergic synapses, we focused on ozanimod, a MS drug recently approved in the United States for clinically isolated syndrome, RRMS, and active SPMS (https://packageinserts.bms.com/pi/pi_zeposia.pdf). Among the multiple beneficial effects obtained during the clinical trials, brain volume loss was significantly reduced compared to treatment with IFN-β-1a both at 12 (SUNBEAM) and 24 (RADIANCE B) months in the ozanimod arms, suggesting a protective effect on brain atrophy in both cohorts [30,45]. In accordance to direct CNS effects exerted by ozanimod, oral therapeutic administration of this drug was shown to ameliorate EAE clinical score in a dose-dependent manner [29]. This result seems to be related only in part to the restriction of autoreactive lymphocyte trafficking, as indicated by the lack of lymphopenia in mice receiving the lowest, but clinically overt effective dose used in the study. Hence, the authors argued that other direct CNS effects exerted by ozanimod may exist [29].

Here, we focused on ozanimod action on EAE synaptic alterations by studying its effects in an ex vivo experimental paradigm. First, we found that ozanimod corrected both the presynaptic and the postsynaptic alterations of the sEPSCs in EAE corticostriatal slices. Next, we investigated the S1P_1_ agonist AUY954 and the S1P_5_ agonist A971432 in ex vivo system and observed that the engagement of S1P_1_ is more efficient in ameliorating the glutamatergic alterations in the EAE striatum, with respect to S1P_5_. Altogether these ex vivo results highlight a specific and fully protective action of ozanimod relatively to AUY954 (no effect on sEPSC frequency) and A971432 (partial effect on sEPSC kinetics and no effect on frequency).

Overall, these observations further strengthen the idea that the complexity of the sphingosine system implies a fine-tuning regulation of the balance between receptor affinity of the agonist and receptor levels. It is still missing a clear picture of potential changes in the expression of S1P receptors in EAE and in MS [46], not only in the different brain area but also at cellular levels. Moreover, the trafficking of S1P receptors and the downstream signaling might be differentially modulated by each drug and by the duration of the treatment [24].

Although further investigations are necessary to define specific intracellular pathways triggered by each S1P receptor modulator, our results further support a neuroprotective effect of S1P-signaling in EAE disease, mediated by an impairment of excitotoxic damage through modulation of S1P_1_ on microglia cells and T lymphocytes. We previously demonstrated that TNF release from activated microglia cells could mimic the glutamatergic alterations observed in EAE striatum [28]. Similarly, we observed that T lymphocytes derived either from the spleen of EAE mice [28] or from blood of RRMS patients during a relapse (chimeric ex vivo MS model), could exert a direct glutamatergic synaptotoxic effect on healthy murine striatal slices [11]. As already mentioned, S1P_1_ appears to be the most expressed in multiple cell types in the CNS, including neurons, astrocytes, and microglia. Astrocytes have been previously demonstrated to respond to several S1P compounds both in vitro [47,48] and in vivo. In particular, Choi and colleagues [24] elegantly demonstrated a clear S1P_1_ involvement in neuroprotection by S1P agonists (Fingolimod and AUY954) in the EAE model, by inducing EAE in conditional null mutants for S1P_1_ in neurons and in astrocytes. These conditional mutants exhibited the predicted pattern of stable disease progression, although clinical signs were attenuated when S1P_1_ was deleted from astrocytes, indicating a non-neuronal and likely astrocyte locus for S1P_1_ signaling [24]. Of note, the involvement of other non-CNS cell lineages expressing S1P_1_, like microglia [49] and endothelial cells [50], which have roles in discrete phases of EAE or MS, have been little explored so far. A neuroprotective effect mediated by S1P_1_ modulation in microglia cells was observed in mice challenged with ischemia and orally treated with AUY954, and in S1P1 knockout mice [51]. Furthermore, the same authors showed that in LPS-stimulated mouse primary microglia transfected with S1P_1_ siRNA, gene expression of proinflammatory mediators such as TNF and IL-1β, but not IL-6, was suppressed [51]. In the present manuscript, we suggest the involvement of these immune cells in the anti-excitotoxic effect mediated by S1P_1_ modulation. By immunofluorescence experiments we showed that bath application of ozanimod (the same condition of electrophysiology) has an anti-inflammatory effect and reduces the expression of the synaptopathic molecules IL-1β and, to a lesser extent, TNF in lesioned areas of EAE striatum, characterized by strong microglia/macrophage activation. In line with this, a global qPCR analysis of EAE striatal slices showed a significant reduction of IL-1β and TNF mRNA after ozanimod treatment together with an up-regulation of the mRNA of the M2-like marker FIZZ1. The anti-inflammatory effect of ozanimod was further supported by in vitro experiments showing that a direct and brief treatment of Th1-activated microglial BV2 cells with ozanimod was sufficient to downregulate the mRNAs of important pro-inflammatory cytokines.

Another potential mechanism responsible of ozanimod anti-inflammatory and neuroprotective activity involves T cells. Here, we showed that pretreatment of EAE T cells with ozanimod abrogates the excitotoxic effects of EAE T cells, likely through an anti-inflammatory action. These results highlight an additional way through which ozanimod could exert its beneficial effect also in MS patients, not only by reducing the infiltration of T cells, but also by modulating synaptotoxic T cells that circulate in the EAE/MS brain. It is worth noting that, by using an ex vivo MS chimeric model, we recently demonstrated that T cells derived from active MS patients exert a direct synaptotoxic effect, similarly to EAE lymphocytes [13]. Therefore, we expect that RRMS lymphocytes treated with ozanimod are less synaptotoxic in ex vivo MS chimeric experiments.

We believe that the central anti-excitotoxic effect exerted by S1P_1_ modulation could have an impact also on clinical disability, as suggested by AUY954 icv delivery in EAE mice. To exclude a S1P_1_-mediated peripheral restraint of T cells during treatment we verified the lack of lymphopenia in mice receiving central infusion of AUY954. These results are in accordance with the identification of non-immunological CNS mechanisms mediated by FTY720 and AUY954 [24].

In conclusion, in this study we show a selective and central action of S1P_1_ modulators in ameliorating EAE glutamatergic synaptopathy and disease. Furthermore, we propose that ozanimod exerts a central anti-inflammatory activity by engaging the S1P_1_ expressed by microglia cells and peripheral T cell infiltrating in the brain. Altogether these data strengthen the relevance of a neuro-immunomodulatory and -protective action of S1P_1_ agonist in MS therapy.

## Figures and Tables

**Figure 1 cells-09-01290-f001:**
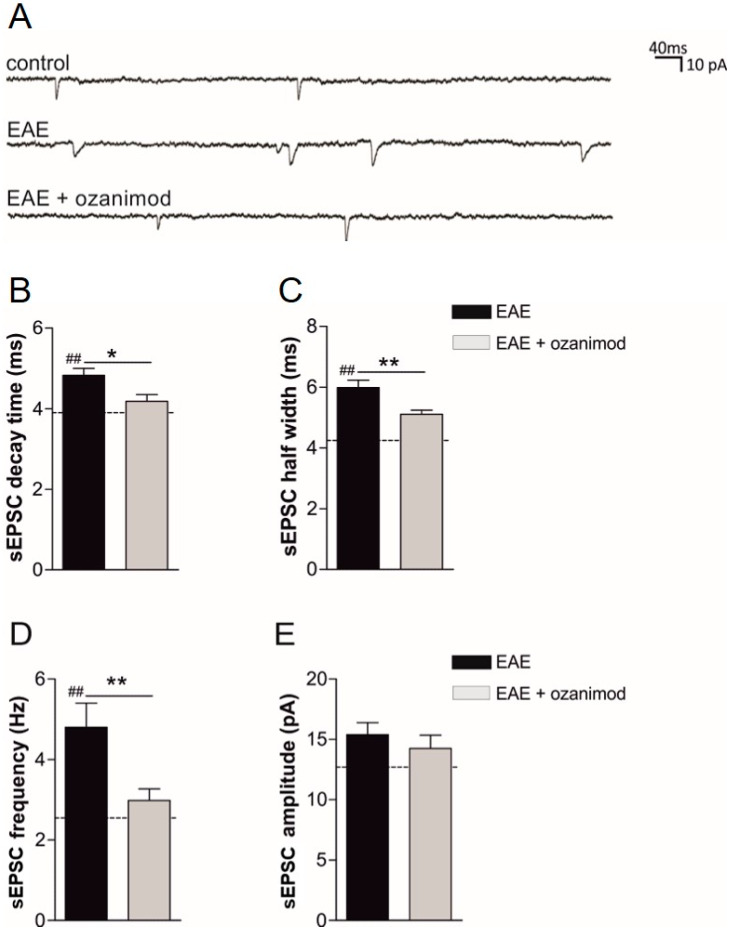
Ex vivo treatment of corticostriatal slices with ozanimod recovers synaptic alterations induced by experimental autoimmune encephalomyelitis (EAE). (**A**) Examples of spontaneous glutamate-mediated excitatory postsynaptic currents (sEPSC) traces recorded from medium spiny neurons (MSNs) in corticostriatal slices in the different experimental conditions (healthy control, EAE, EAE + ozanimod). Bath incubation corticostriatal slices with ozanimod (1000 nM, 2 h) recovers EAE-induced alterations of glutamatergic transmission, in terms of decay time (**B**), half width (**C**), and frequency (**D**). sEPSC amplitude is not affected by ozanimod treatment (**E**). Dotted lines refer to healthy mouse values. Data are expressed as mean ± SEM. Unpaired *t*-test, * *p* < 0.05; ** *p* < 0.01 EAE vs. EAE + ozanimod; ^##^
*p* < 0.01 EAE vs. healthy mice.

**Figure 2 cells-09-01290-f002:**
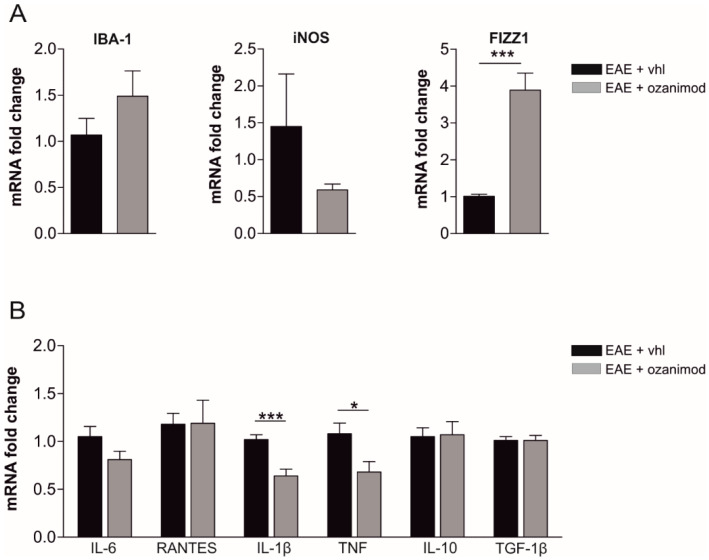
Ex vivo treatment of EAE corticostriatal slices with ozanimod modulates inflammatory markers related to microglial activation and lowers IL-1β and TNF mRNA levels. (**A**) qPCR quantification of microglial markers from EAE striatal slices incubated with ozanimod (1000 nM,2 h) shows a significant upregulation of the M2 marker FIZZ1 (resistin like alpha, Retnla) without changing the expression of iNOS (inducible nitric oxide synthetase) and IBA1 (binding adaptor molecule 1). (**B**) qPCR quantification of cytokines from EAE striatal slices incubated with ozanimod (1000 nM, 2 h) shows a significant downregulation of IL-1β and TNF mRNAs. All data are expressed as mean ± SEM and as fold change of EAE vehicle samples. Unpaired *t*-test, * *p* < 0.05; *** *p* < 0.001.

**Figure 3 cells-09-01290-f003:**
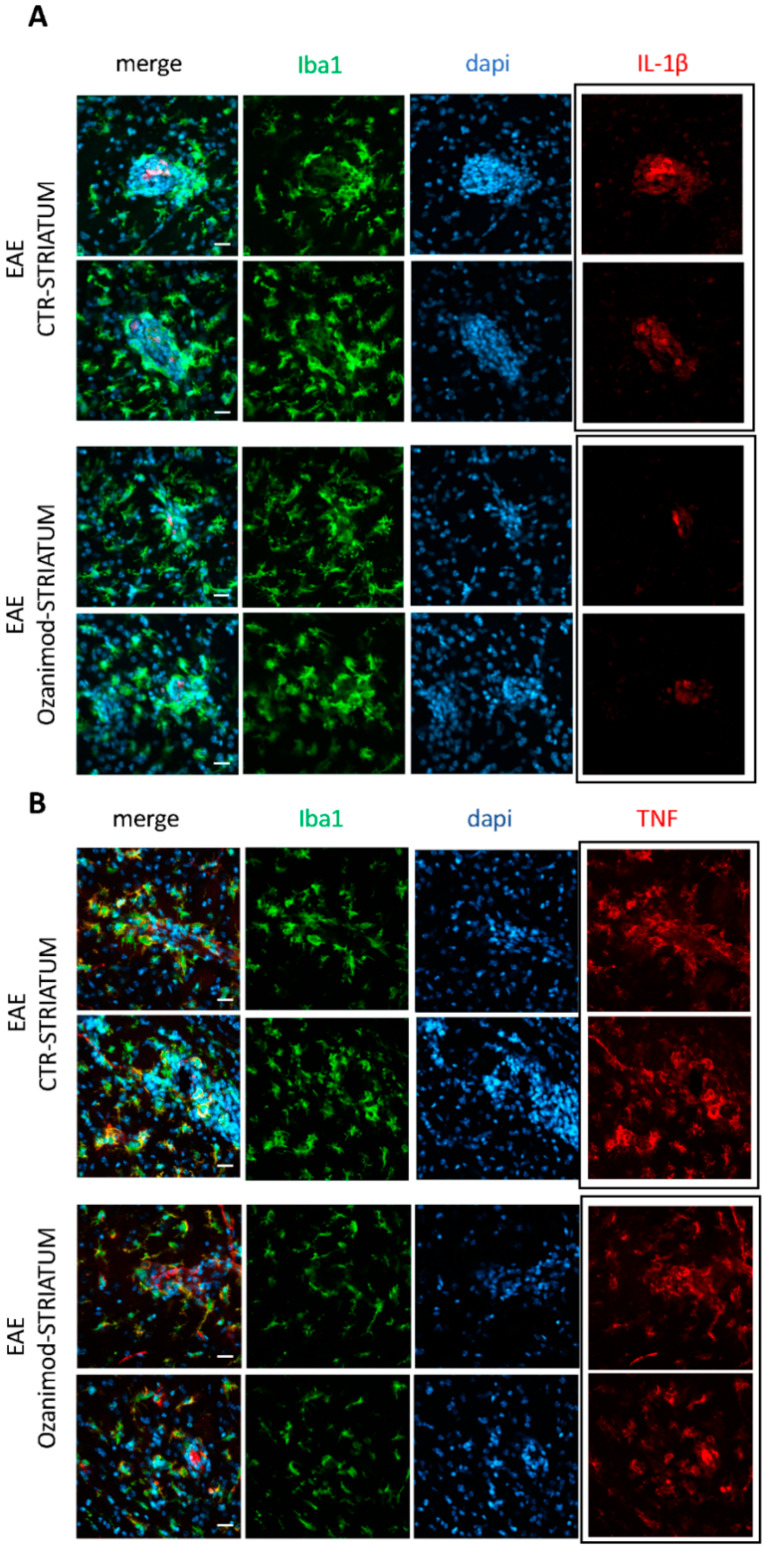
Ozanimod attenuates IL-1β immunolabelling with negligible effect on TNF in EAE striatal slices. Confocal images of EAE striatal slices incubated with vehicle or ozanimod (1000 nM, 2 h) stained for IBA1 (green), cell nuclei (cyan), IL-1 β (red in **A**), and TNF (red in **B**) show that ozanimod treatment leads to a significantly milder expression of IL-1 β within lesioned area, highlighted by IBA1 and dapi staining, with minor effect on TNF. Scale bars: 20 µm.

**Figure 4 cells-09-01290-f004:**
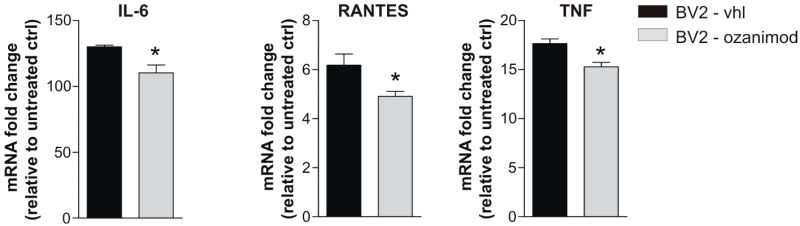
In vitro treatment of BV2 cell line with ozanimod modulates mRNA levels of pro-inflammatory cytokines. qPCR experiments performed on BV2 microglial cells activated by Th1 Mix for 2 h and incubated with ozanimod (1000 nM, 1 h pretreatment) or vehicle (DMSO) show a downregulation of IL-6, TNF and RANTES mRNAs. All data are expressed as mean ± SEM and as fold change of untreated controls. Unpaired *t*-test, * *p* < 0.05.

**Figure 5 cells-09-01290-f005:**
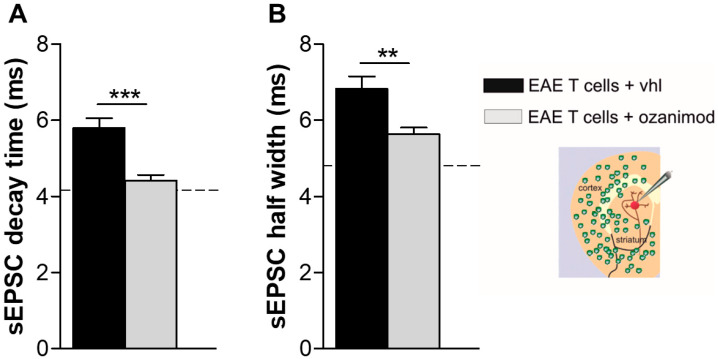
In vitro treatment of EAE T cells with ozanimod abolishes T cell synaptotoxicity. (**A**,**B**) The enhancement of sEPSC decay time (**A**) and sEPSC half width (**B**),typically induced by EAE lymphocytes, was significantly reduced by in vitro treatment of EAE T cells with ozanimod (1000 nM). Dotted lines refer to control condition (control T cells). Data are presented as ± SEM. Unpaired *t*-test, ** *p* < 0.01; *** *p* < 0.001.

**Figure 6 cells-09-01290-f006:**
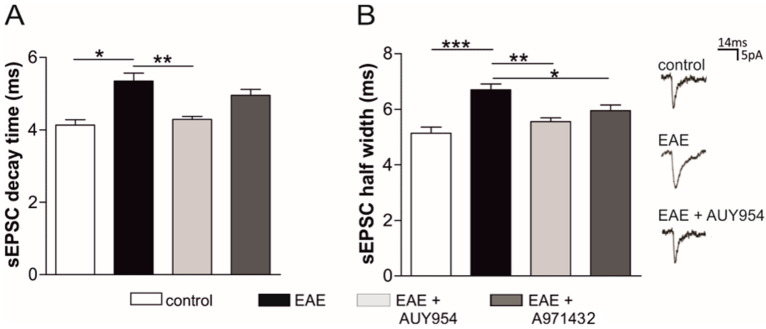
Ex vivo treatment with S1P_1_ and S1P_5_ agonists ameliorates glutamatergic dysfunction in EAE striatal slices. Whole-cell patch clamp recordings from MSNs show that the kinetics of the glutamatergic currents, decay time (**A**) and half width (**B**), were increased in EAE striatum and were completely rescued after 2 h incubation with AUY954 (300 nM, S1P_1_ specific). A971432 (200 nM, S1P5 specific) only partially rescued the EAE sEPSC kinetics. On the right, representative peaks of electrophysiological recordings in the different experimental conditions are shown. Data are presented as mean ± SEM. ANOVA, * *p* < 0.05; ** *p* < 0.01; *** *p* < 0.001.

**Figure 7 cells-09-01290-f007:**
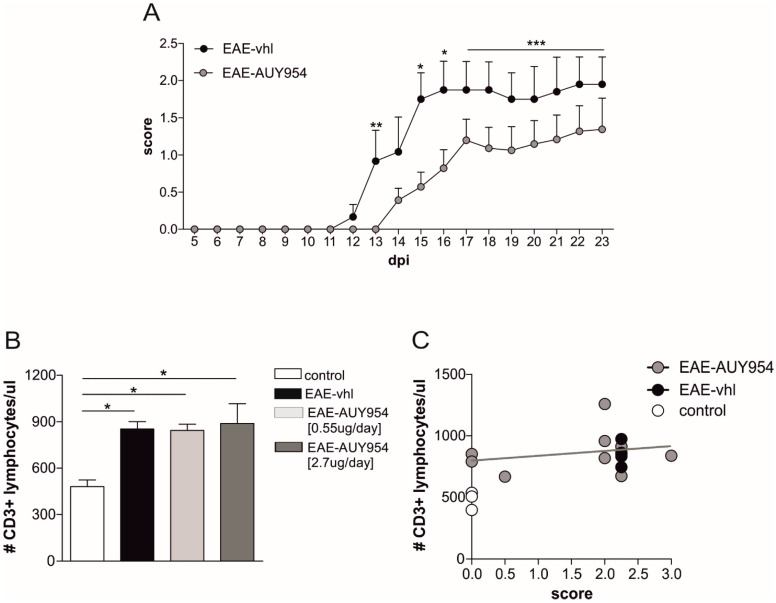
AUY954 icv treatment ameliorates EAE clinical disability without affecting T cell absolute count. (**A**) The graph shows representative clinical course of EAE mice treated for four weeks with two different AUY954 dosages (2.7 µg/day and 0.55 µg/day) or vehicle, preventively delivered by icv infusion. AUY954 icv treatment significantly ameliorated EAE disease progression. Mann-Whitney test on day 13, 15, 16, and from 17 to 23 days post immunization (dpi) by cumulating the data, * *p* < 0.05; *** p* < 0.01; *** *p* < 0.001). (**B**) CD3+ lymphocytes were counted in the peripheral blood of EAE mice (21 dpi) receiving vehicle (vhl) or different AUY954 dosages, 2.7 µg/day and 0.55 µg/day. No significant reduction was observed at any dosage in comparison to EAE-vhl mice. Conversely, T cell count was significantly less in healthy mice in comparison to other EAE groups. Data are presented as mean ± SEM; ANOVA Tukey’s *p* < 0.05. (**C**) Statistical analysis between clinical score and T cell count shows the lack of correlation between these two parameters in AUY954-EAE mice (r^2^ = 0.06).

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
