# Peer review of "Central Modulation of Selective Sphingosine-1-Phosphate Receptor 1 Ameliorates Experimental Multiple Sclerosis"

_cells, 2020, doi:10.3390/cells9051290_

Round 1

Reviewer 1 Report

The manuscript by Musella et al. is a very interesting paper aimed at demonstrating the pivotal role of the inhibition of some sphingosine-1-phosphate receptors 1 in the control of experimental multiple sclerosis.

The manuscript is in general well written and the experiments accurately designed, performed, described and commented. There are, however, the following concerns that should be clarified:

  1. In some Figures, namely the Figures 1A and 5A-B, it is reported a dotted line indicating the values recorded in control cells, which are those present in brain tissue slices of healthy animals. The statistical analysis has been performed comparing results from EAE cells with these control (untreated) cells, whereas results related to the use of ozanimod only concerned the effect of this treatment in EAE cells in comparison to untreated EAE cells. I think that the Authors should also show or report the effect of ozanimod in cells from healthy control, to point out the major involvement and role of S1P receptors in the examined pathological condition.
  2. In relation to the Fig. 1, panels A, B and C, there is a symbol ## that should be related to any statistical evaluation, but no explanation in this sense is reported in the legend. Please, clarify.

Minor points:

- Among the keyword, there is the term “citokine”, which should be “cytokines” since in the paper several of them have them have been investigated; as well S1P1 and S1P5 should be followed by receptors. The same is valid when these terms are used in the text of the article. Alternatively, the Authors may choose to adopt an abbreviation such as S1P1/5R.

- In the text and in the figure 7A it is reported the term “dpi” without any explanation.

- At page 2, line 79, the drug BAF should be indicated as BAF312

- At page 3, line 126, the dose used for AUY954 are already reported at the line 100 of the same page. Therefore, it may be eliminated at the line 126

- At the page 3, again, lines 128-129, the Authors report that slices were obtained by the use of a vibratome, without any other explanation, whereas at pag. 4, in the paragraph concerning electrophysiology, they give more details about the same instrument. I think that the details about vibratome should be included only in the paragraph about “ex vivo experiments”

- Please, include a correct explanation for the abbreviation “ACSF” (pages 4 and 5)

- There are some few errors for English (i.e. at page 14, lines 441-443: “lowets” instead of “lowest” or “may exists” instead of “may exist”). Therefore, check the text.

Author Response

REVIEWER 1

The manuscript by Musella et al. is a very interesting paper aimed at demonstrating the pivotal role of the inhibition of some sphingosine-1-phosphate receptors 1 in the control of experimental multiple sclerosis.

R: We thank the reviewer for the positive comment.

The manuscript is in general well written and the experiments accurately designed, performed, described and commented. There are, however, the following concerns that should be clarified:

In some Figures, namely the Figures 1A and 5A-B, it is reported a dotted line indicating the values recorded in control cells, which are those present in brain tissue slices of healthy animals. The statistical analysis has been performed comparing results from EAE cells with these control (untreated) cells, whereas results related to the use of ozanimod only concerned the effect of this treatment in EAE cells in comparison to untreated EAE cells. I think that the Authors should also show or report the effect of ozanimod in cells from healthy control, to point out the major involvement and role of S1P receptors in the examined pathological condition.

R: We agree with the reviewer about the importance of evaluating the effect of ozanimod in physiological condition. However, in our previous paper we performed similar experiments with fingolimod (Rossi et al., 2012) and we did not observe any effect of the drug on synaptic transmission in corticostriatal slices of healthy mice. Furthermore, based on our experience, in control conditions it is difficult to evaluate a further reduction of the of sEPSC kinetic parameters following a pharmacological treatment. On the other hand, in pathological conditions, characterized by the alterations of these parameters, it is possible to appreciate a recovery of sEPSC kinetics.

In relation to the Fig. 1, panels A, B and C, there is a symbol ## that should be related to any statistical evaluation, but no explanation in this sense is reported in the legend. Please, clarify.

R: We added this missing information in the figure legend.

Minor points:

  • Among the keyword, there is the term “citokine”, which should be “cytokines” since in the paper several of them have them have been investigated; as well S1P1 and S1P5 should be followed by receptors. The same is valid when these terms are used in the text of the article. Alternatively, the Authors may choose to adopt an abbreviation such as S1P1/5R.

R: In agreement with the reviewer, we changed the word “cytokine” with “cytokines”. Regarding the S1P1/5 receptors, we applied the nomenclature used for these receptors based on the recommendation to write the receptor number as a subscript, without adding “R” or “receptor”.

  • In the text and in the figure 7A it is reported the term “dpi” without any explanation.

R: We explained the acronym “dpi” in the method section and in the figure legend (see pg 4 and 14).

  • At page 2, line 79, the drug BAF should be indicated as BAF312

R: We corrected the text

  • At page 3, line 126, the dose used for AUY954 are already reported at the line 100 of the same page. Therefore, it may be eliminated at the line 126

R: In agreement with the reviewer, we changed the text.

  • At the page 3, again, lines 128-129, the Authors report that slices were obtained by the use of a vibratome, without any other explanation, whereas at pag. 4, in the paragraph concerning electrophysiology, they give more details about the same instrument. I think that the details about vibratome should be included only in the paragraph about “ex vivo experiments”

    R: The text has been corrected accordingly.

  • Please, include a correct explanation for the abbreviation “ACSF” (pages 4 and 5)

    R: We provided the correct explanation for the “ACSF” abbreviation in the text (see pg 4).

  • There are some few errors for English (i.e. at page 14, lines 441-443: “lowets” instead of “lowest” or “may exists” instead of “may exist”). Therefore, check the text.

R: We apologize for the mistakes, we corrected the text and made another round of text editing.

Reference:

Rossi et al., 2012. Oral fingolimod rescues the functional deficits of synapses in experimental autoimmune encephalomyelitis. Br J Pharmacol. 2012;165(4):861-9.

Submission Date

20 April 2020

Date of this review

23 Apr 2020 12:06:47

Reviewer 2 Report

In the present manuscript, Musella et al have investigated the effect of the anti-inflammatory/neuroprotective effects of ozanimod (RPC1063) on a mouse model of multiple sclerosis (MS). This S1P1/5 receptor agonist was recently approved in the US for the treatment of MS. The investigation uses several techniques suitable for this kind of studies and seems to be carried out carefully. I have a few questions that should be addressed:

  • In Figure 1, I think it would be better to show the traces (now Fig. 1C) as Fig. 1A. Furthermore, the data in Fig 1B probably is statistically significant for EAE + ozanimod compared with the data for healthy mice. Thus, it is not proper to say that the values are similar to healthy mice.
  • In Fig. 2 and 4 mRNA data is used to show changes in inflammatory markers and cytokines. mRNA data often is clear indications of changes in protein levels, but it would be proper to point out that the data may not necessarily relate to changes in protein levels.
  • The immunostaining for TNF in Fig. 3 is not convincingly showing a difference between control and ozanimod treatment.
  • In the legend to Fig. 5, the Authors state that “sEPSC kinetics (in terms of decay time and half width) typically induced by EAE 327 lymphocytes, was completely prevented”. I would rather say significantly reduced, as was also used in the Results text (line 317).
  • To help the reader, it would be good to mention the n-numbers also in the Figure legends.
  • On line 429 the Authors use FTY720, but through the text they use fingolimod, please change to fingolimod to be consistent
  • On line 436, the sentence”…, trials, brain volume loss was significantly reduced compared to IFN-β-1a…” is missing something. Should it state compared to treatment with IFN-β-1a?
  • On line 441, the word “but” should probably not be there as it makes the sentence strange.

Author Response

In the present manuscript, Musella et al have investigated the effect of the anti-inflammatory/neuroprotective effects of ozanimod (RPC1063) on a mouse model of multiple sclerosis (MS). This S1P1/5 receptor agonist was recently approved in the US for the treatment of MS. The investigation uses several techniques suitable for this kind of studies and seems to be carried out carefully.

R: We thank the reviewer for the positive comments.

I have a few questions that should be addressed:

In Figure 1, I think it would be better to show the traces (now Fig. 1C) as Fig. 1A. Furthermore, the data in Fig 1B probably is statistically significant for EAE + ozanimod compared with the data

R: The figure has been corrected accordingly. Moreover, the statistical analysis between control condition and EAE+ ozanimod, showing no differences between the two groups, was added in the text, in figure1 and its relative legend (see pg 7)

In Fig. 2 and 4 mRNA data is used to show changes in inflammatory markers and cytokines. mRNA data often is clear indications of changes in protein levels, but it would be proper to point out that the data may not necessarily relate to changes in protein levels. The immunostaining for TNF in Fig. 3 is not convincingly showing a difference between control and ozanimod treatment.

R: We thank the reviewer for the observation. We have changed the text to clarify this issue (see pg 9 and 16).

In the legend to Fig. 5, the Authors state that “sEPSC kinetics (in terms of decay time and half width) typically induced by EAE 327 lymphocytes, was completely prevented”. I would rather say significantly reduced, as was also used in the Results text (line 317).

R: We changed the text.

To help the reader, it would be good to mention the n-numbers also in the Figure legends.

R: We added this information in the figure legends.

On line 429 the Authors use FTY720, but through the text they use fingolimod, please change to fingolimod to be consistent

R: In agreement with the reviewer, we changed the text

On line 436, the sentence”…, trials, brain volume loss was significantly reduced compared to IFN-β-1a…” is missing something. Should it state compared to treatment with IFN-β-1a?

R: We changed the text as suggested by the reviewer.

On line 441, the word “but” should probably not be there as it makes the sentence strange

R: We apologize for the mistake, we corrected the text.

Round 2

Reviewer 1 Report

I think that the paper has appropriately been revised by the Authors, who fulfilled all requests by the reviewers so that, in may opinion, it can be accepted for publication